# Turkish Adaptation of the Ghosting Questionnaire and Its Impact on Relationship Satisfaction: Serial Mediation Effects of Negative Affect and Loneliness

**DOI:** 10.3390/ejihpe15050071

**Published:** 2025-05-07

**Authors:** Mehmet Özalp, Waqar Husain, Kamile Gamze Yaman, Achraf Ammar, Khaled Trabelsi, Seithikurippu R. Pandi-Perumal, Haitham Jahrami

**Affiliations:** 1Department of Social Sciences and Humanities, Naval Academy, National Defense University, İstanbul P.O. Box 34942, Türkiye; mehmet.ozalp@msu.edu.tr; 2Department of Humanities, COMSATS University Islamabad, Islamabad Campus, Park Road, Islamabad 45550, Pakistan; drsukoon@gmail.com; 3Department of Educational Sciences, Faculty of Education, Marmara University, İstanbul P.O. Box 34852, Türkiye; gamze.alcekic@marmara.edu.tr; 4Department of Training and Movement Science, Institute of Sport Science, Johannes Gutenberg-University Mainz, 55099 Mainz, Germany; acammar@uni-mainz.de; 5Research Laboratory, Molecular Bases of Human Pathology, LR19ES13, Faculty of Medicine of Sfax, University of Sfax, Sfax 3000, Tunisia; 6High Institute of Sport and Physical Education of Sfax, University of Sfax, Sfax 3000, Tunisia; trabelsikhaled@gmail.com; 7Department of Movement Sciences and Sports Training, School of Sport Science, The University of Jordan, Amman P.O. Box 11942, Jordan; 8Centre for Research and Development, Chandigarh University, Mohali 140413, Punjab, India; pandiperumal2023@gmail.com; 9Division of Research and Development, Lovely Professional University, Phagwara 144411, Punjab, India; 10Government Hospitals, Manama P.O. Box 12, Bahrain; 11Department of Psychiatry, College of Medicine and Health Sciences, Arabian Gulf University, Manama P.O. Box 26671, Bahrain

**Keywords:** ghosting, scale, adaptation, measurement invariance, item response theory, mediation, negative affect, loneliness, relationship satisfaction

## Abstract

Background: Ghosting is a prevalent phenomenon in contemporary relationships, impacting all individuals involved. A measurement tool, the Ghosting Questionnaire, has recently been developed to assess experiences of ghosting. The objective of this study was to adapt the Ghosting Questionnaire for use in Turkish. Methods: The adaptation process involved applying confirmatory factor analysis, measurement invariance analysis by gender, and item response theory, as well as reliability, criterion-related validity, and predictive validity analyses. Results: Confirmatory factor analysis affirmed the unidimensional, eight-item structure of the Ghosting Questionnaire. Measurement invariance analysis by gender indicated that the scale assesses the same constructs for both males and females. The results from the item response theory analysis demonstrated that the Ghosting Questionnaire possesses robust discriminatory power. Reliability coefficients indicated a high level of internal consistency for the scale. Additionally, ghosting was found to have significant correlations with various variables, including personality traits, positive affect, negative affect, loneliness, and relationship satisfaction. Notably, our findings revealed that negative affect and loneliness serve as serial mediators in the relationship between ghosting and relationship satisfaction. Conclusions: The analyses confirm that the Ghosting Questionnaire is a measurement tool with strong psychometric properties.

## 1. Introduction

Owing to developments in technology in recent years and the impact of social media, which has become an integral part of daily life, interaction patterns are diversifying, relationship dynamics are changing, and new forms of behavior are observed in relationships ([37]; [45]). These behaviors can occur through online and offline channels in different social contexts, such as romantic relationships, friendships, and work environments ([18]; [28]; [30]; [51]; [54]). As one of the behaviors in this context, ghosting is widely seen in relationships today and affects everyone who is a part of the relationship ([20]).

As a type of relationship dissolution strategy ([34]; [41]), ghosting, one of the ways in which people end a relationship today, is the sudden cessation of communication with a person for no apparent reason or the sudden ignoring of a person with whom one is interacting ([30]). It may occur in romantic relationships ([6]; [19]; [34]), friendships ([10]; [18]; [54]), mobile flirting relationships ([33]; [49]), familial relationships ([10]; [48]), professional relationships ([10]; [14]; [51]), or therapeutic relationships ([17]). Even if technological possibilities play an important role ([10]; [14]), ghosting can be experienced not only through online channels or social media platforms ([12]; [33]; [49]) but also in offline environments ([30]; [32]; [43]; [48]). An individual may experience ghosting not only during adolescence ([15]) but also during emerging/young adulthood ([18]; [33]; [53]) or adulthood ([39], [40]). There are no significant gender-specific differences between males and females, so both can experience ghosting ([6]; [19]; [40]). Moreover, ghosting can occur not only once but also repeatedly ([35]). The fact that ghosting, which can be encountered in different types of relationships and various age groups, has become a subject of research in many countries ([10]; [20]; [26], [27]; [29]; [33]; [40]) reveals that ghosting is an important and universal factor affecting relationships.

The reflection of ghosting on the relationship leads to various consequences for both ([21]; [52]) the person who is the ghoster and the person who is the ghostee ([34]). The ghoster may feel guilty or relieved, whereas the ghostee may feel sad or hurt ([14]; [21]). In addition, pioneering studies ([15]; [18]; [39]; [41]; [48]) have noted that ghosting can have many psychological consequences, such as depression, anxiety, awkwardness, self-criticism, self-doubt, hopelessness, anger, loneliness, helplessness, and social avoidance. On the other hand, sometimes, ghosting can also result in positive outcomes, such as resilience and seeing it as an opportunity for growth by assessing oneself for possible future relationships ([48]). In summary, ghosting is a new behavioral concept that is seen in relationships today and has various effects and consequences. Further research is needed to understand the impact of this concept on relationships and its links with other variables in depth. In this context, a measurement tool has recently been developed to measure ghosting experiences ([28]).

This new measurement tool, which is the first instrument to comprehensively measure ghosting ([28]), enables the measurement of ghosting experiences functionally and has already been used in studies on the subject ([20]; [42]; [47]). The original form of the scale is in English, and adaptation studies have been carried out in Urdu ([26]) and Arabic ([27]). Therefore, the main purpose of our study is to adapt the mentioned measurement tool ([28]) into Turkish and enable ghosting-themed studies to be conducted in Türkiye. In this respect, within the scope of this research, we first test the validity and reliability of the Ghosting Questionnaire. After this, we examine the relationships between ghosting and various variables (personality traits, positive affect, negative affect, loneliness, and relationship satisfaction) to ensure criterion-related validity. Finally, we investigate the role of negative affect and loneliness as serial mediators in the relationship between ghosting and relationship satisfaction to ensure predictive validity. In this context, investigations into the relationships between ghosting and personality traits, positive affect, negative affect, loneliness, and relationship satisfaction are planned.

This study fills a gap in the literature by providing a Turkish adaptation of the Ghosting Questionnaire, a tool designed to measure ghosting experiences, and exploring its psychometric properties and relationships with negative affect, loneliness, and relationship satisfaction. Despite ghosting’s prevalence in contemporary relationships and its impact on emotional well-being, our understanding of its measurement and effects in diverse cultures is limited. By adapting and validating this scale for a Turkish sample, this study enhances research on ghosting and its cross-cultural understanding. Additionally, investigating the mediation effects of negative affect and loneliness offers insights into the psychological mechanisms linking ghosting to reduced relationship satisfaction, highlighting the significance of this research in relationship science and psychometric studies.

## 2. Materials and Methods

### 2.1. Procedure

As part of the process of adapting the Ghosting Questionnaire in Turkish, the scale was first translated into Turkish. The Ghosting Questionnaire was translated into Turkish following the linguistic equivalence procedures outlined by Brislin ([7], [8]). Initially, two academic language specialists, fluent in both English and Turkish, translated the questionnaire from English to Turkish (M.Ö., K.G.Y.). Following this initial translation, the Turkish versions were reviewed and consolidated into a single, coherent form. Subsequently, a different language specialist (paid translator) translated this consolidated Turkish version back into English to verify its accuracy and consistency with the original. The back-translated English version was then compared with the original questionnaire, and any discrepancies were addressed. Finally, the Turkish versions were evaluated by experts in psychology and linguistics to ensure cultural relevance and appropriateness. This meticulous process aimed to preserve the integrity and meaning of the original questionnaire while ensuring its suitability for the Turkish context. Importantly, none of the items were modified during the translation process.

Exploratory and confirmatory factor analyses of the scale were subsequently conducted, and a measurement invariance analysis was performed according to gender. After this, the scale was analyzed via item–total correlations and item response theory. Reliability analyses will be conducted. Within the scope of criterion-related validity analysis, the relationships between ghosting and personality traits, positive affect, negative affect, loneliness, and relationship satisfaction were examined. Finally, for predictive validity, a structural equation model addressing the relationships among ghosting, negative affect, loneliness, and relationship satisfaction was tested.

### 2.2. Participants

For this study, data were collected twice (phase I and phase II) in different periods. For the analyses to be conducted in phase I, data were collected from 454 people (female = 357, 78.6%; male = 97, 21.4%) living in 35 of the 81 cities in Türkiye via an online survey. The average age of the participants was 22.31 years (SD = 4.97 years, 18–53 years). Among the participants, 5.3% were high school graduates, 1.1% were associate degree graduates, 78.4% were undergraduate students, 10.8% were bachelor’s degree graduates, 3.5% were master’s degree graduates, and 0.9% were PhD graduates. Regarding relationship status, 57.5% of the participants stated that they did not have a partner, 4% stated that they were dating, 29.5% stated that they had a partner, 0.7% stated that they were engaged, 7% stated that they were married, 0.2% stated that they were divorced, and 1.1% chose the option “other”. Concerning employment status, 75.3% of the participants stated that they were not employed, 10.6% stated that they were employed part-time, 12.8% stated that they were employed full-time, and 1.3% selected the option “other”. With respect to socioeconomic status (SES), 1.5% of the participants had very low SES, 10.4% had low SES, 76.9% had middle SES, 10.1% had high SES, and 1.1% had very high SES.

For the analyses planned to be carried out in this phase (phase II) of the research, data were collected from 309 people living in 38 cities in Türkiye via an online survey. The average age of the participants was 27.57 years (SD = 7.92 years, 18–60 years), and there was a balanced gender distribution (female = 155, 50.2%; male = 154, 49.8%). The participants were 9.4% high school graduates, 4.2% associate degree graduates, 28.2% undergraduate students, 44% bachelor’s degree graduates, 11.9% master’s degree graduates, and 2.3% PhD graduates. Regarding relationship status, 41.1% of the participants stated that they did not have a partner, 4.5% stated that they were dating, 24.9% stated that they had a partner, 4.2% stated that they were engaged, 24.6% stated that they were married, and 0.6% stated that they were divorced. For employment status, 38.2% of the participants reported that they were not employed, 4.9% reported that they had a part-time job, 54.7% reported that they had a full-time job, and 2.3% selected the option “other”. In addition, 2.3%, 14.2%, 75.1%, and 8.4% of the participants reported having very low, low, middle, and high socioeconomic status, respectively.

### 2.3. Data Collection

Data were collected through Google Forms via online platforms. Before data collection, the participants were informed about the content of the study and notified that they could withdraw from the study at any time, and their consent was obtained.

### 2.4. Measures

Ghosting Questionnaire. The scale developed by [28] ([28]) has a unidimensional structure and consists of eight items (e.g., “Their reply/response messages are delayed”). A five-point Likert scale (1 = never, 5 = always) is used to answer the items. Scores on the scale range from 8 to 40. Higher scores indicate a greater experience of ghosting (for more detailed information, see ([28])).

Ten-Item Personality Inventory (TIPI). This 10-item short form was developed to measure the Big Five personality traits ([24]). These Big Five personality traits are extraversion, agreeableness, conscientiousness, emotional stability, and openness to experience. There are 2 items representing each trait in the scale. For each trait, one of these 2 items is reverse-scored. All items are rated on a seven-point Likert scale (1 = disagree strongly, 7 = agree strongly). There is no total score for the scale, and increasing scores for each trait indicate that the trait has increased. In the Turkish adaptation of the TIPI used in this study ([3]), the five-factor structure of the scale was shown to have acceptable fit indices: χ^2^/df = 2.20; CFI = 0.93; GFI = 0.95; IFI = 0.93; SRMR = 0.042; RMSEA = 0.037.

Positive and Negative Affect Schedule (PANAS) scale. This scale consists of 20 items, half of which are related to positive affect (items 1, 3, 5, 9, 10, 12, 14, 16, 17, and 19) and the other half to negative affect (items 2, 4, 6, 7, 8, 11, 13, 15, 18, and 20). There are no reverse items, and no total score is calculated ([50]). The scores of the items related to the 2 dimensions (positive affect and negative affect) in the scale, which are answered on a five-point Likert scale (1 = very slightly or not at all, 5 = extremely), are summed, and 2 separate scores are obtained. For both dimensions, higher scores indicate greater positive and negative affect. A Turkish adaptation study ([23]) revealed that the PANAS is a valid and reliable measurement tool in a Turkish sample.

UCLA Loneliness Scale (ULS-8). For this 8-item unidimensional scale, items 3 and 6 are reverse-scored. Scores on a four-point Likert scale (1 = not at all appropriate, 4 = completely appropriate) range from 8 to 32, with higher scores indicating greater loneliness ([25]). The Turkish adaptation study ([16]) revealed that the unidimensional structure of the scale consisting of 8 items showed acceptable fit indices: χ^2^ (18, N = 553) = 56.03, GFI = 0.97; CFI = 0.94; NFI = 0.92; IFI = 0.94; RMSEA = 0.066.

Relationship assessment scale. This scale, which was developed to measure satisfaction with relationships with relatives and friends, has a unidimensional structure and consists of 7 items. Items 4 and 7 on the five-point Likert scale (1 = not good at all, 5 = very good) are reverse-scored items. The scores that can be obtained from the scale vary between 7 and 35, and high scores indicate that the satisfaction level of an individual with their relationships with relatives and friends is high ([46]). The Turkish adaptation study ([13]) revealed that the unidimensional structure of the scale consisting of 7 items showed acceptable fit indices: χ^2^ (13, N = 336) = 52.87, GFI = 0.95; CFI = 0.97; NFI = 0.95; IFI = 0.97; RMSEA = 0.069.

### 2.5. Ethics

All studies in this research were conducted in accordance with the principles of the Helsinki Declaration of 1964, which was last revised in October 2024. Furthermore, approval was obtained from the Research and Publication Ethics Committee of Marmara University Institute of Educational Sciences.

### 2.6. Statistical Analyses

#### 2.6.1. Descriptive Statistics

Means, standard deviations (SDs), skewness, and kurtosis were calculated for all variables to summarize the data distribution and assess baseline characteristics. Normality assumptions were evaluated using the Shapiro–Wilk test, as it is more sensitive for small to moderate sample sizes; the results indicated acceptable normality.

#### 2.6.2. Two-Sample Validation Approach

This research employed a rigorous two-sample approach to scale validation. Two independent samples were used, one for exploratory analyses (total 454) and another for confirmatory analyses (total 309), allowing for a more robust validation process.

##### Phase I (Exploratory Analyses; N = 454)

We performed an exploratory factor analysis (EFA) to examine the dimensionality of the scale and identify the underlying factor structure. The EFA was conducted using maximum likelihood extraction without rotation, as we aimed to explore the natural factor structure without imposing any constraints on the factor solution.

##### Phase II (Confirmatory Analyses; N = 309)

The confirmatory factor analysis (CFA) of the scale was performed via maximum likelihood estimation in the AMOS program. The goodness-of-fit index (GFI), normed fit index (NFI), incremental fit index (IFI), Tucker–Lewis index (TLI), comparative fit index (CFI), standardized root mean square residual (SRMR), and root mean square error of approximation (RMSEA) were used to evaluate the model fit.

To assess the convergent and discriminant validity of the scale, we computed the average variance extracted (AVE) and heterotrait–monotrait (HTMT) ratio of correlations. The AVE indicates the variance captured by the construct relative to measurement errors, while the HTMT ratio compares correlations between items measuring different constructs to those measuring the same construct, providing a stringent assessment of discriminant validity.

#### 2.6.3. Measurement Invariance

A measurement invariance analysis by gender was also conducted via the AMOS program. The item–total correlations of the scale were examined.

#### 2.6.4. Item Response Theory (IRT)

The discriminatory power of the scale items was examined via the graded response model (GRM) via item response theory (IRT) in the Stata Version MP 17 program.

#### 2.6.5. Reliability Analyses

Several reliability coefficients, such as Cronbach’s alpha (α), McDonald’s omega (ω), and Guttman’s lambda (λ6), were calculated.

#### 2.6.6. Criterion-Related Validity

The correlations between ghosting and the Big Five personality traits, positive affect, negative affect, loneliness, and relationship satisfaction were calculated via Pearson’s correlation coefficients.

#### 2.6.7. Other Analyses

We then carried out two-step structural equation modeling (SEM) in the AMOS program, using maximum likelihood estimation for parameter estimation. In this context, we first tested whether the measurement model was confirmed and then tested the structural model ([2]; [31]). We also used the item parceling method ([5]; [38]) for the unidimensional scales in the model that we tested. A mediation analysis was conducted to examine the possible mediating roles of negative affect and loneliness in the relationship between ghosting and relationship satisfaction. Additionally, we used bootstrap testing to determine whether the indirect effects were significant ([44]).

Using data from phase I and phase II combined, an independent-samples *t*-test was conducted to compare the ghosting scores between female (n = 512) and male (n = 251) participants. The test was used to determine if there was a statistically significant difference between the mean scores of the two groups.

In all analyses, a *p*-value of less than 0.05 was considered statistically significant.

## 3. Results

Using a sample of 454 individuals (phase I) for the EFA revealed a unidimensional structure with one significant factor explaining the variance across the eight measured variables (Item 1–Item 8). The factor loadings ranged from moderate to strong (0.41–0.84), with Item 6 showing the strongest association with the factor (0.84), followed by Item 5 (0.78) and Item 3 (0.77). Item 4 demonstrated the weakest loading (0.41) and highest uniqueness value (0.83). The assumption checks supported the appropriateness of the factor analysis. Bartlett’s test of sphericity was significant (χ^2^ (28) = 1677.83, *p* < 0.001), indicating sufficient correlations among the variables. The overall Kaiser–Meyer–Olkin (KMO) measure of sampling adequacy was excellent at 0.89, with all individual variables showing strong MSA values (0.87–0.93), confirming the suitability of the data for factor analysis.

Using the second sample of 309 individuals (phase II), the CFA demonstrated an acceptable model fit for the Turkish version of the eight-item original form of the Ghosting Questionnaire: χ^2^ (20, N = 454) = 127.73, *p* < 0.001; GFI = 0.93; NFI = 0.92; IFI = 0.93; TLI = 0.90; CFI = 0.93; SRMR = 0.045; RMSEA 0.075; AVE = 0.50; HTMT = 1.0. In other words, the unidimensional, eight-item structure of the Ghosting Questionnaire was confirmed. The factor loadings ranged between 0.410 and 0.837. The factor loadings, item–total correlations, and descriptive statistics for the Ghosting Questionnaire are presented in Table 1.

After confirming the structure of the scale, a measurement invariance analysis was performed according to gender. The invariance can be assessed by the value of ∆CFI being equal to or less than 0.01 and the value of ∆TLI being less than 0.05 ([11]; [36]). The ∆CFI and ∆TLI values were acceptable, as shown in Table 2. These results show that the scale demonstrated configural, metric, and scalar invariance across gender groups and measured the same construct in males and females.

Following the measurement invariance analysis by gender, an IRT analysis was performed. The results of the IRT analysis are presented in Table 3 and Figure 1. Based on these results, the discrimination parameter (*a*) values of six out of eight items in the scale are very high, whereas the discrimination parameter (*a*) values of two items are moderate ([4]). These findings show that the discriminative power of the Ghosting Questionnaire is reasonably good.

The reliability coefficients of the Ghosting Questionnaire (separately for phase I and phase II) were analyzed with Cronbach’s alpha, McDonald’s omega, and Guttmann’s lambda. The reliability coefficients of the Ghosting Questionnaire calculated in phase I and phase II are presented in Table 4. According to Table 4, the Cronbach’s alpha (α = 0.876–0.879), McDonald’s omega (ω = 0.884–0.886), and Guttmann’s lambda (λ6 = 0.878–0.875) values regarding the reliability of the Ghosting Questionnaire are highly acceptable.

The correlations between ghosting and the Big Five personality traits, positive affect, negative affect, loneliness, and relationship satisfaction are presented in Table 5. Accordingly, while there was a significant negative relationship between ghosting and the conscientiousness (*r* = −0.225, *p* < 0.001) and emotional stability (*r* = −0.207, *p* < 0.001) personality traits, there was no significant relationship between ghosting and the extraversion (*r* = −0.065, *p* > 0.05), agreeableness (*r* = −0.078, *p* > 0.05), or openness to experience (*r* = −0.009, *p* > 0.05) personality traits. Moreover, it was concluded that there was a significant negative relationship between ghosting and positive affect (*r* = −0.136, *p* < 0.05) and relationship satisfaction (*r* = −0.281, *p* < 0.001), whereas there was a significant positive relationship between ghosting and negative affect (*r* = 0.287, *p* < 0.001) and loneliness (*r* = 0.268, *p* < 0.001).

The measurement model consisted of four latent variables (ghosting, negative affect, loneliness, and relationship satisfaction) and nine observed variables. The findings of the measurement model showed that the model and the data fit each other satisfactorily: χ^2^ (21, N = 309) = 49.64, *p* < 0.001; χ^2^/df = 2.36; GFI = 0.96; NFI = 0.96; RFI = 0.93; IFI = 0.97; TLI = 0.96; CFI = 0.97; SRMR = 0.035; RMSEA = 0.067. After the measurement model was confirmed, the structural model was analyzed. We first tested the full mediation model (Model I), in which the serial mediating role of negative affect and loneliness in the relationship between ghosting and relationship satisfaction was examined. Although the results of Model I showed that the model had acceptable fit indices—χ^2^ (22, N = 309) = 57.62; χ^2^/df = 2.61; GFI = 0.95; NFI = 0.95; RFI = 0.93; IFI = 0.97; TLI = 0.95; CFI = 0.97; SRMR = 0.047; RMSEA = 0.073; AIC = 103.625; ECVI = 0.336—the path from negative affect to relationship satisfaction was found to be insignificant in this model. Therefore, this path was removed, and the model (Model II) was analyzed again. The results of Model II indicated that the model had acceptable fit indices: χ^2^ (23, N = 309) = 59.67; χ^2^/df = 2.59; GFI = 0.95; NFI = 0.95; RFI = 0.93; IFI = 0.97; TLI = 0.95; CFI = 0.97; SRMR = 0.051; RMSEA = 0.072; AIC = 103.670; ECVI = 0.337. After the full mediation model regarding the serial mediating role of negative affect and loneliness in the relationship between ghosting and relationship satisfaction was tested, the partial mediation model (Model III) was tested. Although the results of Model III demonstrated that the model had acceptable fit indices—χ^2^ (21, N = 309) = 49.64; χ^2^/df = 2.36; GFI = 0.96; NFI = 0.96; RFI = 0.93; IFI = 0.97; TLI = 0.96; CFI = 0.97; SRMR = 0.035; RMSEA = 0.067; AIC = 97.645; ECVI = 0.317—the links from negative affect to relationship satisfaction (*p* = 0.430) and from ghosting to loneliness (*p* = 0.093) were not significant in this model. For this reason, the path from negative affect to relationship satisfaction was first removed, and the model (Model IV) was reanalyzed. Although the results of Model IV showed that the model had acceptable fit indices—χ^2^ (22, N = 309) = 50.26; χ^2^/df = 2.28; GFI = 0.96; NFI = 0.96; RFI = 0.93; IFI = 0.97; TLI = 0.96; CFI = 0.97; SRMR = 0.036; RMSEA = 0.065; AIC = 96.267; ECVI = 0.313—the path from ghosting to loneliness was found to be insignificant in this model. For this reason, this pathway was removed, and the model (Model V) was reanalyzed. The results of Model V indicated that the model had acceptable fit indices—χ^2^ (23, N = 309) = 52.76; χ^2^/df = 2.29; GFI = 0.96; NFI = 0.96; RFI = 0.93; IFI = 0.97; TLI = 0.96; CFI = 0.97; SRMR = 0.043; RMSEA = 0.065; AIC = 96.761; ECVI = 0.314—and all paths in the model also became significant. Because a model with lower AIC and ECVI coefficients is considered superior ([1]; [9]), the partial mediation model was preferred over the full mediation model. In addition, unlike the other partial mediation model options, all paths in Model V were significant, so Model V was selected as the final model and is presented in Figure 2.

Finally, the statistical significance of the mediating variables was tested. This was performed by bootstrapping 5000 samples. The indirect effects of the serial mediation model are shown in Table 6. The mediating role of negative affect in the relationship between ghosting and loneliness was confirmed by bootstrapping (*β* = −0.282, *p* < 0.001, 95% CI = 0.063, 0.187). On the other hand, negative affect was associated with relationship satisfaction through loneliness (*β* = −0.313, *p* < 0.001, 95% CI = −0.421, −0.218). Finally, the serial mediating role of negative affect and loneliness in the relationship between ghosting and relationship satisfaction was confirmed (*β* = −0.089, *p* < 0.001, 95% CI = −0.153, −0.043).

The analysis of the independent-samples *t*-test revealed virtually identical mean ghosting scores between females (M = 18.13, SD = 6.25) and males (M = 18.12, SD = 6.29). The independent-samples I-test indicated no statistically significant difference between the groups (*t*(761) = 0.037, *p* = 0.975, Cohen’s d = 0.00). The mean difference was minimal at 0.01 points, with a standard error of difference of 0.48. These results demonstrate the complete absence of gender-based performance differences in the ghosting measure.

## 4. Discussion

Ghosting, which has started to be seen more intensely in relationships in recent years, affects many areas of life, especially relationships, as a new behavioral pattern. This study aimed to examine the psychometric properties of the Ghosting Questionnaire in a Turkish sample and to investigate the relationship between ghosting and various variables (personality traits, positive affect, negative affect, loneliness, and relationship satisfaction).

The two-phase analytical approach in this study enhances the methodological rigor of our findings. The exploratory factor analysis with the first sample revealed a unidimensional structure, confirmed by confirmatory factor analysis with an independent sample. Using separate samples for EFA and CFA mitigates potential issues of chance capitalization ([31]). The consistency of the findings across two independent samples increases the confidence in the stability of the factor structure.

The convergent validity of the scale was supported by an acceptable AVE value of 0.50, meeting the minimum threshold, while the HTM value of 1.0 indicated complete discriminant validity between the construct and other variables in the measurement model.

As in the original English version ([28]) of the Ghosting Questionnaire and its Urdu ([26]) and Arabic forms ([27]), the unidimensional structure of the scale consisting of eight items was confirmed in the Turkish version as a result of the confirmatory factor analysis. The results of the measurement invariance analysis by gender revealed that the Ghosting Questionnaire measured the same construct in males and females. Analyses using item response theory showed that the discrimination power of all items in the Ghosting Questionnaire was above the acceptable limit value ([4]), and 75% of the items (6 items) had very high discrimination power. In other words, the Ghosting Questionnaire can distinguish between individuals with and without ghosting experience quite well. At the same time, the reliability coefficients calculated in two groups of data collected at different periods show that the reliability of the Ghosting Questionnaire is at a high level.

The relationships between ghosting and the Big Five personality traits, positive affect, negative affect, loneliness, and relationship satisfaction were examined. Whereas a significant negative relationship was found between ghosting and the personality traits of conscientiousness and emotional stability, no significant relationship was found between ghosting and other personality traits (extraversion, agreeableness, openness to experience). In a study conducted on this subject ([6]), it was concluded that there was a significant negative relationship between ghosting and emotional stability personality traits, as in our study. Unlike our findings, in the same study, a significant negative relationship was found between ghosting and the personality trait of agreeableness, whereas no significant relationship was found between ghosting and the personality trait of conscientiousness. Our results also demonstrated significant negative relationships between ghosting and positive affect and relationship satisfaction, as well as significant positive relationships between ghosting and negative affect and loneliness. Whereas the results of a limited number of studies on these variables overlap with our findings, there are also points of difference. Similarly to our results, studies ([41]; [53]) have reported that individuals exposed to ghosting experience many positive and negative emotions. In another study ([32]), the results between ghosting and negative affect were found in parallel with our findings. On the other hand, whereas one study ([39]) reported that there was no relationship between ghosting and loneliness, another study ([53]) revealed that individuals exposed to ghosting may start to feel loneliness as a result of ghosting.

Finally, we examined the serial mediating role of negative affect and loneliness in the relationship between ghosting and relationship satisfaction. Our findings indicated that negative affect and loneliness have a serial mediator role in the relationship between ghosting and relationship satisfaction. In other words, people who are more exposed to ghosting are more likely to experience negative affect. Individuals who experience more negative affect are more likely to feel loneliness. Increased feelings of loneliness lead to lower relationship satisfaction.

The serial mediation pathway identified in this study—ghosting → negative affect → loneliness → relationship satisfaction—can be clarified through emotional reasoning ([22]). Emotions shape cognitive appraisals and amplify psychological distress. For instance, individuals experiencing negative affect after ghosting may engage in distorted interpretations, such as overgeneralization (“No one values me”), catastrophizing (“I’ll never connect with others”), or personalization (“I caused this rejection”). These maladaptive cognitions intensify feelings of alienation and withdrawal, exacerbating loneliness. Loneliness then diminishes relationship satisfaction, reinforcing negative self-schemas and emotional distress. Gangemi et al.’s (2021) framework highlights how emotional reasoning connects affective states and cognitive biases, explaining the observed mediation. Future research could measure these cognitive distortions to clarify their role in the ghosting–loneliness–satisfaction cycle, especially in cross-cultural contexts with differing norms around emotional expression.

The psychometric properties of the Ghosting Questionnaire ([26], [27]; [28]) indicate its use in academic research and clinical practice. It can enhance psychotherapeutic contexts for individuals exhibiting ghosting behaviors or affected by such experiences. This tool may help clinicians to understand the patterns, motivations, and emotions associated with ghosting. By integrating the Ghosting Questionnaire into assessments, practitioners can gain insights into clients’ personality traits, emotional states, and relational dynamics, facilitating tailored interventions for issues like emotional instability, negative affect, and loneliness. This approach could improve therapeutic outcomes by providing a framework to explore and address the psychological consequences of ghosting, fostering healthier relationship behaviors.

Ghosting, often linked to romantic relationships, also affects friendships and professional contexts, leading to abrupt communication breakdowns. This can cause feelings of rejection and betrayal, as these relationships are built on trust and respect. The mental health impacts are significant, contributing to anxiety, loneliness, and depression. Addressing ghosting across all relationship types is vital in understanding its effects on social and emotional well-being. Thus, future research should consider ghosting beyond romantic contexts to gain insights into its broader mental health implications and inform interventions to mitigate its negative consequences.

To summarize, although the findings of our study largely overlap with the results of a limited number of studies on the subject, more research is needed to understand the connections between the constructs that are likely to be related to ghosting in depth. In this respect, the results of our study show that the Ghosting Questionnaire is a valid and reliable measurement tool in the Turkish sample and can be used with peace of mind in future studies and contribute to the related field.

### Limitations

This study provides valuable insights into the psychological consequences of ghosting but has limitations. The data in the study were collected using self-reported scales, which may introduce some errors due to subjectivity. It is also difficult to reach causal conclusions due to the cross-sectional design of the study. In this regard, there is a need to conduct longitudinal and experimental studies to reveal causal relationships and effects. The sample size, while adequate, was not extensive, which may affect the generalizability of the findings. Additionally, the sample was disproportionately female; thus, future research should aim for larger and more balanced samples to validate these findings. Despite these constraints, this study is a significant step in understanding ghosting, a pervasive phenomenon with profound emotional and relational repercussions.

We suggest the following areas to be explored. First, longitudinal designs could track ghosting experiences over time to establish causal pathways between ghosting, negative affect, loneliness, and relationship satisfaction, particularly during critical life transitions (e.g., emerging adulthood). Second, experimental studies could manipulate ghosting scenarios (e.g., simulated digital interactions) to isolate its immediate emotional and behavioral consequences. Third, comparative cross-cultural investigations are needed to explore how cultural norms (e.g., collectivism vs. individualism) shape perceptions and outcomes of ghosting, building on this Turkish adaptation. Fourth, mixed-methods approaches (e.g., qualitative interviews paired with physiological stress measures) could uncover the motivations behind ghosting behaviors and their somatic impacts. Finally, intervention studies could evaluate strategies (e.g., resilience training, digital etiquette programs) to mitigate ghosting-related distress. These directions would deepen theoretical insights while informing practical solutions for individuals and communities navigating modern relational challenges.

## 5. Conclusions

We examined the relatively new concept of “ghosting” by analyzing two different samples from which we collected data at different periods. The analyses indicated that the Ghosting Questionnaire is a measurement tool that exhibits strong psychometric properties. Moreover, our cross-sectional examinations have shown that ghosting may lead to greater levels of negative affect and loneliness and lower levels of relationship satisfaction. In this context, there is a need to conduct more experimental and longitudinal studies to better understand the concept of ghosting and to confirm its relationships with the relevant variables in this research.

## Figures and Tables

**Figure 1 ejihpe-15-00071-f001:**
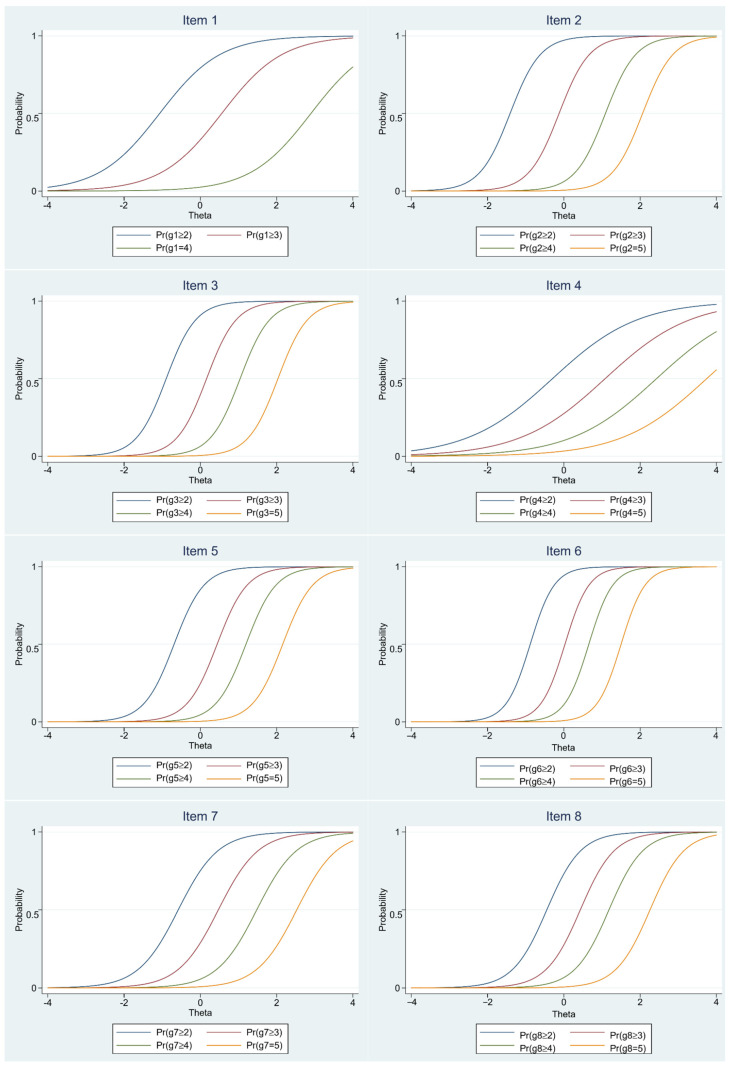
Item characteristics curve for the Ghosting Questionnaire.

**Figure 2 ejihpe-15-00071-f002:**
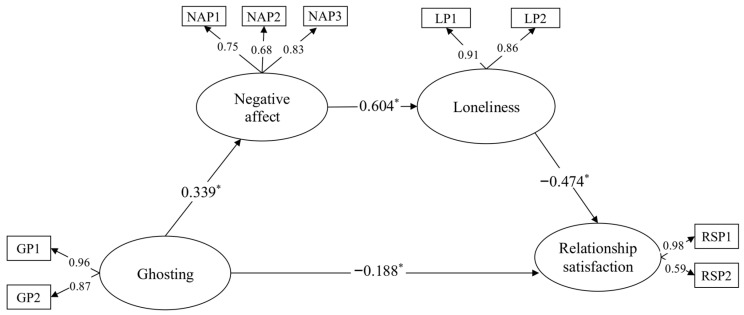
Structural equation modeling for the serial mediation model. Note. * *p* < 0.001; GP1 and GP2: parcels of ghosting; NAP1, NAP2, and NAP3: parcels of negative affect; LP1 and LP2: parcels of loneliness; RSP1 and RSP2: parcels of relationship satisfaction.

**Table 1 ejihpe-15-00071-t001:** Statistics on the items of the Ghosting Questionnaire.

	Factor Loading	Corrected Item–Total Correlation	*M* (SD)	Skewness	Kurtosis
Item 1	0.531 *	0.508	2.16 (0.86)	0.126	−0.896
Item 2	0.761 *	0.698	2.65 (1.01)	0.308	−0.284
Item 3	0.771 *	0.721	2.46 (1.12)	0.415	−0.627
Item 4	0.410 *	0.388	2.04 (1.16)	0.941	−0.039
Item 5	0.778 *	0.717	2.27 (1.11)	0.653	−0.367
Item 6	0.837 *	0.772	2.65 (1.27)	0.308	−1.007
Item 7	0.675 *	0.639	2.19 (1.10)	0.640	−0.396
Item 8	0.727 *	0.674	2.21 (1.17)	0.642	−0.639

* *p* < 0.001.

**Table 2 ejihpe-15-00071-t002:** Fit indices of gender invariance.

Invariance	χ^2^	*df*	IFI	TLI	CFI	RMSEA	SRMR	∆CFI	∆TLI
Configural invariance	151.866 *	40	0.937	0.906	0.933	0.079	0.046	–	–
Metric invariance	161.228 *	47	0.932	0.919	0.932	0.073	0.047	0.001	0.013
Scalar invariance	174.461 *	54	0.928	0.925	0.928	0.070	0.047	0.004	0.006

* *p* < 0.001.

**Table 3 ejihpe-15-00071-t003:** IRT results for the Ghosting Questionnaire.

Item	*a* Coefficient	SE	Confidence Interval	*z*	*p* > |*z*|
Item 1	1.26	0.13	1.01–1.51	9.84	0.001
Item 2	2.51	0.21	2.09–2.91	11.97	0.001
Item 3	2.56	0.21	2.14–2.98	11.99	0.001
Item 4	0.89	0.11	0.67–1.12	7.89	0.001
Item 5	2.54	0.22	2.12–2.97	11.73	0.001
Item 6	3.18	0.28	2.64–3.72	11.49	0.001
Item 7	1.90	0.17	1.57–2.23	11.26	0.001
Item 8	2.23	0.19	1.85–2.61	11.48	0.001

**Table 4 ejihpe-15-00071-t004:** Reliability results of the Ghosting Questionnaire.

	Phase I (N = 454)	Phase II (N = 309)
Cronbach’s alpha	0.876	0.879
McDonald’s omega	0.884	0.886
Gutmann’s lambda	0.878	0.875

**Table 5 ejihpe-15-00071-t005:** Descriptive statistics and correlations with Ghosting Questionnaire (GQ).

			Correlation with GQ	
Variable	Mean	SD	*r*	*p*
Ghosting	17.40	5.831	-	-
Big Five personality traits				
Extraversion	10.32	2.598	−0.065	0.252
Agreeableness	10.71	2.179	−0.078	0.169
Conscientiousness	11.64	2.136	−0.225	<0.001 *
Emotional stability	9.72	2.451	−0.207	<0.001 *
Openness to experience	9.46	2.238	−0.009	0.874
Positive affect	35.33	7.598	−0.136	0.016 *
Negative affect	21.09	6.393	0.287	<0.001 *
Loneliness	12.23	3.510	0.268	<0.001 *
Relationship satisfaction	26.06	4.149	−0.281	<0.001 *

* *p* < 0.05.

**Table 6 ejihpe-15-00071-t006:** Indirect effects of the serial mediation model.

Indirect Effect	Coefficient	95% CI	
		LL	UL
Ghosting → negative affect → loneliness	0.120	0.063	0.187
Ghosting → negative affect → loneliness → relationship satisfaction	−0.089	−0.153	−0.043
Negative affect → loneliness → relationship satisfaction	−0.313	−0.421	−0.218

Note. CI, confidence interval; LL, lower limit; UL, upper limit.

## Data Availability

Data are available on request.

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
