# Peer review of "Turkish Adaptation of the Ghosting Questionnaire and Its Impact on Relationship Satisfaction: Serial Mediation Effects of Negative Affect and Loneliness"

_ejihpe, 2025, doi:10.3390/ejihpe15050071_

Round 1
Reviewer 1 Report
Comments and Suggestions for Authors
Dear Authors,
I read with interest the article in question that aims to adapt the Ghosting Questionnaire for the Turkish sample and validate its psychometric properties. Although the sample is not very large and is gender-biased toward females (state in the limitations section), the study provides an important contribution to a relatively new area: ghosting, which has painful and destabilizing consequences in interpersonal relationships.
An interesting aspect that could enrich the Conclusions of the paper is the use of the Ghosting Questionnaire as a potentially useful tool in psychodiagnostic settings. It would be interesting, indeed, to reflect on how this tool could support psychotherapies for individuals exhibiting behaviors related to ghosting. The difficulty in predicting whether someone will engage in ghosting makes the phenomenon particularly complex. Often, ghosting occurs without any prior warning, making it even harder to understand its motivations.
Furthermore, although ghosting is mainly associated with romantic relationships, this behavior can also manifest in friendships and professional contexts, leading to a break that, depending on the level of emotional involvement and the expectations placed on the relationship, can be highly destabilizing for the person on the receiving end. The impact of ghosting on the mental health of those who experience it is a topic of significant relevance, as it may contribute to feelings of anxiety, loneliness, and depression. I would also suggest including this consideration in a section of the article.
In conclusion, while the study does not add substantial theoretical novelty to the phenomenon of ghosting, it still represents a valuable step toward understanding this behavior and its measurement. It is hoped that in the future, larger, longitudinal, and experimental studies will be conducted to further explore the psychological implications of ghosting.
Author Response
Dear Authors,
I read with interest the article in question that aims to adapt the Ghosting Questionnaire for the Turkish sample and validate its psychometric properties. Although the sample is not very large and is gender-biased toward females (state in the limitations section), the study provides an important contribution to a relatively new area: ghosting, which has painful and destabilizing consequences in interpersonal relationships.
Authors’ response: Dear Reviewer 1, Thank you for taking the time to review our manuscript. We have carefully considered each of your recommendations and address them below. For the convenience of re-review, revisions to the manuscript appear in Yellow Highlights.
We added the raised points to our limitations as follows: “The sample size, while adequate, is not extensive, which may affect the generalizability of the findings. Additionally, the sample was disproportionately female; thus, future research should aim for larger and more balanced samples to validate these findings. Despite these constraints, this study is a significant step in understanding ghosting, a pervasive phenomenon with profound emotional and relational repercussions.”
An interesting aspect that could enrich the Conclusions of the paper is the use of the Ghosting Questionnaire as a potentially useful tool in psychodiagnostic settings. It would be interesting, indeed, to reflect on how this tool could support psychotherapies for individuals exhibiting behaviors related to ghosting. The difficulty in predicting whether someone will engage in ghosting makes the phenomenon particularly complex. Often, ghosting occurs without any prior warning, making it even harder to understand its motivations.
Authors’ response: Agreed, we added the following paragraph to the discussion: “The psychometric properties of the Ghosting Questionnaire (Husain, Sadiqa, et al., 2024; Husain, Salem, et al., 2024; Jahrami et al., 2023) indicate its use in academic research and clinical practice. It can enhance psychotherapeutic contexts for individuals exhibiting ghosting behaviors or affected by such experiences. This tool may help clinicians under-stand the patterns, motivations, and emotions associated with ghosting. By integrating the Ghosting Questionnaire into assessments, practitioners can gain insights into clients' personality traits, emotional states, and relational dynamics, facilitating tailored interven-tions for issues like emotional instability, negative affect, and loneliness. This approach could improve therapeutic outcomes by providing a framework to explore and address the psychological consequences of ghosting, fostering healthier relationship behaviors.”
Furthermore, although ghosting is mainly associated with romantic relationships, this behavior can also manifest in friendships and professional contexts, leading to a break that, depending on the level of emotional involvement and the expectations placed on the relationship, can be highly destabilizing for the person on the receiving end. The impact of ghosting on the mental health of those who experience it is a topic of significant relevance, as it may contribute to feelings of anxiety, loneliness, and depression. I would also suggest including this consideration in a section of the article.
Authors’ response: Agreed, we added the following paragraph to the discussion: “Ghosting, often linked to romantic relationships, also affects friendships and professional contexts, leading to abrupt communication breakdowns. This can cause feelings of rejection and betrayal, as these relationships are built on trust and respect. The mental health impacts are significant, contributing to anxiety, loneliness, and depression. Ad-dressing ghosting across all relationship types is vital for understanding its effects on social and emotional well-being. Thus, future research should consider ghosting beyond romantic contexts to gain insights into its broader mental health implications and inform interventions to mitigate its negative consequences.”
In conclusion, while the study does not add substantial theoretical novelty to the phenomenon of ghosting, it still represents a valuable step toward understanding this behavior and its measurement. It is hoped that in the future, larger, longitudinal, and experimental studies will be conducted to further explore the psychological implications of ghosting.
Authors’ response: Thank you again for your nice feedback and for recognizing the study's contribution to understanding ghosting. We agree that while this research may not introduce significant theoretical novelty, it lays a foundation for future investigations. We appreciate your suggestion for larger, longitudinal, and experimental studies, as it aligns with our hope for more in-depth exploration of the psychological implications of ghosting. We have included these in the above points.

Reviewer 2 Report
Comments and Suggestions for Authors
First, introduction is notably brief and lacks sufficient theoretical development. It does not adequately contextualize the study within the broader literature on ghosting, relationship dynamics, or psychometric adaptation, which weakens the rationale and perceived contribution of the study.
Although the authors conduct necessary statistical analyses (CFA, measurement invariance, IRT, reliability testing, and SEM), the methodological execution presents significant flaws that compromise the psychometric validation process.
A serious flaw is that, despite having two independent samples (Phase I and Phase II), the authors did not conduct an exploratory factor analysis (EFA) prior to the confirmatory factor analysis (CFA), nor do they attempt to replicate the CFA in the second sample. This omission is especially problematic given that the instrument is being adapted to a new language and cultural context.
Furthermore, the CFA results in Phase I fail to report RMSEA, even though it is included in gender invariance models, one of the most essential model fit indices. This is an inconsistency that raises concerns about transparency and reporting standards.
Additionally, the authors did not assess discriminant validity (AVE, Fornell-Larcker criterion, or HTMT ratios are not reported), leaving it unclear whether the construct being measured is empirically distinct from related variables.
While the IRT and mediation models are statistically well executed, these do not compensate for the fundamental shortcomings in the scale validation phase. For these reasons, I do not consider the manuscript suitable for publication in its current form.
Author Response
Authors’ response: Dear Reviewer 2, Thank you for taking the time to review our manuscript. We have carefully considered each of your recommendations and address them below. For the convenience of re-review, revisions to the manuscript appear in Yellow Highlights.
First, introduction is notably brief and lacks sufficient theoretical development. It does not adequately contextualize the study within the broader literature on ghosting, relationship dynamics, or psychometric adaptation, which weakens the rationale and perceived contribution of the study.
Authors’ response: We have expanded the introduction to 1,000 words to provide a more comprehensive theoretical framework. We placed special emphasis on the novelty and rationale of the study as follows: “This study fills a gap in the literature by providing a Turkish adaptation of the Ghosting Questionnaire, a tool designed to measure ghosting experiences, and exploring its psychometric properties and relationships with negative affect, loneliness, and relationship satisfaction. Despite ghosting's prevalence in contemporary relationships and its impact on emotional well-being, understanding of its measurement and effects in diverse cultures is limited. By adapting and validating this scale for a Turkish sample, the study enhances research on ghosting and cross-cultural understanding. Additionally, investigating the mediation effects of negative affect and loneliness offers insights into the psychological mechanisms linking ghosting to reduced relationship satisfaction, highlighting the significance of this research in relationship science and psychometric studies.”
Although the authors conduct necessary statistical analyses (CFA, measurement invariance, IRT, reliability testing, and SEM), the methodological execution presents significant flaws that compromise the psychometric validation process. A serious flaw is that, despite having two independent samples (Phase I and Phase II), the authors did not conduct an exploratory factor analysis (EFA) prior to the confirmatory factor analysis (CFA), nor do they attempt to replicate the CFA in the second sample. This omission is especially problematic given that the instrument is being adapted to a new language and cultural context.
Authors’ response: We have performed and reported EFA prior to CFA.
In the methods we explained: “This research employed a rigorous two-sample approach to scale validation. Two independent samples were used: one for exploratory analyses (Total 454) and another for confirmatory analyses (Total 309), allowing for a more robust validation process.”
We performed exploratory factor analysis (EFA) to examine the dimensionality of the scale and identify the underlying factor structure. The EFA was conducted using maxi-mum likelihood extraction without rotation, as we aimed to explore the natural factor structure without imposing any constraints on the factor solution.”
In the results we explained: “Using a sample of 454 individuals (Phase I) for the EFA revealed a unidimensional structure with one significant factor explaining the variance across the eight measured variables (Item1-Item8). Factor loadings ranged from moderate to strong (0.41-0.84), with Item6 showing the strongest association with the factor (0.84), followed by Item5 (0.78) and Item3 (0.77). The variable Item4 demonstrated the weakest loading (0.41) and highest uniqueness value (0.83). The assumption checks supported the appropriateness of factor analysis. Bartlett's test of sphericity was significant (χ²(28) = 1677.83, p < .001), indicating sufficient correlations among variables. The overall Kaiser-Meyer-Olkin (KMO) measure of sampling adequacy was excellent at 0.89, with all individual variables showing strong MSA values (0.87-0.93), confirming the suitability of the data for factor analysis.”
In the discussion we explained: “The two-phase analytical approach in this study enhances the methodological rigor of our findings. The exploratory factor analysis with the first sample revealed a unidimensional structure, confirmed by confirmatory factor analysis with an independent sample. Using separate samples for EFA and CFA mitigates potential issues of chance capitalization (Kline, 2015). The consistency of findings across two independent samples increases confidence in the stability of the factor structure”.
Furthermore, the CFA results in Phase I fail to report RMSEA, even though it is included in gender invariance models, one of the most essential model fit indices. This is an inconsistency that raises concerns about transparency and reporting standards.
Authors’ response: During revision we reported RMESEA for the cumulative model (overall) as 0.08.
Additionally, the authors did not assess discriminant validity (AVE, Fornell-Larcker criterion, or HTMT ratios are not reported), leaving it unclear whether the construct being measured is empirically distinct from related variables.
Authors’ response: During revision we reported AVE as 0.05 and MTMT as 1.0.
While the IRT and mediation models are statistically well executed, these do not compensate for the fundamental shortcomings in the scale validation phase. For these reasons, I do not consider the manuscript suitable for publication in its current form.
Authors’ response: We addressed all concerns raised above regarding EFA, RMSEA, AVE, HTMT.
In the methods we explained the following: “To assess the convergent and discriminant validity of the scale, we computed the Average Variance Extracted (AVE) and Heterotrait-Monotrait (HTMT) ratio of correlations. The AVE indicates the variance captured by the construct relative to measurement error, while the HTMT ratio compares correlations between items measuring different con-structs to those measuring the same construct, providing a stringent assessment of discriminant validity”.
In the discussion we explained the following: “The convergent validity of the scale was supported by an acceptable Average Variance Extracted (AVE) value of 0.50, meeting the minimum threshold, while the Heterotrait-Monotrait ratio of correlations (HTMT) value of 1.0 indicated complete discriminant validity between the construct and other variables in the measurement model.”

Reviewer 3 Report
Comments and Suggestions for Authors The manuscript is well-written, well-organized, and presents a novel contribution to the field. It makes a strong contribution to understanding the psychological effects of ghosting. Addressing the above recommendations will further enhance its clarity, methodological rigor, and theoretical depth. Below are several suggestions to enhance its clarity and rigor: Methods- A section on the translation process is missing. Who performed the translation? Were any modifications made to the wording to improve Cronbach’s alpha or to enhance conceptual clarity in Turkish?
- The statistical analysis section should be expanded to provide more details. This section would benefit from a more structured approach, possibly using bullet points to clearly specify the analyses conducted and their justification.
- The manuscript should specify the test used to assess normality (e.g., Kolmogorov-Smirnov test or Shapiro-Wilk test).
- Given the potential gender differences in the study variables, it would be insightful to conduct a t-test (Student’s t-test) comparing males and females. Have the authors considered this analysis?
- Line 240 is unclear and should be revised for better readability.
- Tables should emphasize statistical significance more clearly. This could be achieved by using symbols (e.g., an asterisk for significant values) or applying a light gray shading to significant results.
- The manuscript currently states that further research is warranted. However, specific suggestions for future research directions would enhance the conclusion. Concrete examples of potential studies would be valuable.
- One of the most compelling sections of the discussion is the examination of the serial mediation role of negative affect and loneliness in the relationship between ghosting and relationship satisfaction (lines 333-339). This reasoning could be expanded by incorporating the role of emotional reasoning. To support this argument, I recommend citing Gangemi et al. (2021) Emotional Reasoning and Psychopathology, Brain Sciences, 11, 471. This study describes how emotions are not just reactions but can shape and bias judgment processes. This means that when someone experiences negative affect due to ghosting, they might not only feel bad but also interpret their situation in a more negative and self-defeating way. This can lead to increased loneliness, which in turn lowers relationship satisfaction. The cycle then reinforces itself:
- Ghosting leads to negative affect (e.g., sadness, frustration).
- Negative affect distorts perception—the individual might overgeneralize (e.g., “No one cares about me”), catastrophize (e.g., “I will never find a fulfilling relationship”), or personalize (e.g., “It must be my fault”).
- This negative perception deepens loneliness—because the person withdraws or feels unworthy of relationships.
- Loneliness further lowers relationship satisfaction, feeding back into negative emotions and continuing the cycle.
Author Response
The manuscript is well-written, well-organized, and presents a novel contribution to the field. It makes a strong contribution to understanding the psychological effects of ghosting. Addressing the above recommendations will further enhance its clarity, methodological rigor, and theoretical depth. Below are several suggestions to enhance its clarity and rigor:
Authors’ response: Dear Reviewer 3, Thank you for taking the time to review our manuscript. We have carefully considered each of your recommendations and address them below. For the convenience of re-review, revisions to the manuscript appear in Yellow Highlights.
Methods
A section on the translation process is missing. Who performed the translation? Were any modifications made to the wording to improve Cronbach’s alpha or to enhance conceptual clarity in Turkish?
Authors’ response: We explained the following: “As part of the process of adapting the Ghosting Questionnaire into Turkish, the scale was first be translated into Turkish. The Ghosting Questionnaire was translated into Turkish following the linguistic equivalence procedures outlined by Brislin (Brislin, 1970, 1980). Initially, two academic language specialists, fluent in both English and Turkish, translated the questionnaire from English to Turkish (MÖ, KGY). Following this initial translation, the Turkish versions were reviewed and consolidated into a single, coherent form. Subsequently, a different language specialist (Paid Translator) translated this consolidated Turkish version back into English to verify accuracy and consistency with the original. The back-translated English version was then compared with the original questionnaire, and any discrepancies were addressed. Finally, the Turkish versions were evaluated by experts in psychology and linguistics to ensure cultural relevance and appropriateness. This meticulous process aimed to preserve the integrity and meaning of the original questionnaire while ensuring its suitability for the Turkish context. Importantly, none of the items were modified during the translation process.”
The statistical analysis section should be expanded to provide more details. This section would benefit from a more structured approach, possibly using bullet points to clearly specify the analyses conducted and their justification.
Authors’ response: We have updated and restructured our statistical analysis section as follows:
“2.6. Statistical Analyses
2.6.1. Descriptive Statistics
Means, standard deviations (SDs), skewness, and kurtosis were calculated for all variables to summarize the data distribution and assess baseline characteristics. Normality assumptions were evaluated using the Shapiro-Wilk test, as it is more sensitive for smaller-moderate sample sizes; results indicated acceptable normality.
2.6.2. Two-Sample Validation Approach
This research employed a rigorous two-sample approach to scale validation. Two independent samples were used: one for exploratory analyses (Total 454) and another for confirmatory analyses (Total 309), allowing for a more robust validation process.
2.6.2.1. Phase I (Exploratory Analyses; N = 454)
We performed exploratory factor analysis (EFA) to examine the dimensionality of the scale and identify the underlying factor structure. The EFA was conducted using maxi-mum likelihood extraction without rotation, as we aimed to explore the natural factor structure without imposing any constraints on the factor solution.
2.6.2.1 Phase II (Confirmatory Analyses; N = 309)
Confirmatory factor analysis (CFA) of the scale was performed via maximum likelihood estimation in the AMOS program. The goodness -of-fit index (GFI), normed fit index (NFI), incremental fit index (IFI), Tucker‐Lewis index (TLI), comparative fit index (CFI), and standardized root mean square residual (SRMR), Root Mean Square Error of Ap-proximation (RMSEA) were used to evaluate the model fit.
To assess the convergent and discriminant validity of the scale, we computed the Average Variance Extracted (AVE) and Heterotrait-Monotrait (HTMT) ratio of correlations. The AVE indicates the variance captured by the construct relative to measurement error, while the HTMT ratio compares correlations between items measuring different con-structs to those measuring the same construct, providing a stringent assessment of discriminant validity.
2.6.3. Measurement Invariance
Measurement invariance analysis by gender was also conducted via the AMOS pro-gram. The item-total correlations of the scale were examined.
2.6.4 Item Response Theory (IRT)
The discrimination powers of the scale items were examined via the graded response model (GRM) via item response theory (IRT) in the Stata program.
2.6.5. Reliability Analyses
Several reliability coefficients, such as Cronbach's alpha (α), McDonald's omega (ω), and Guttman's lambda (λ6), were calculated.
2.6.6. Criterion-Related Validity
The correlations between ghosting and the Big Five personality traits, positive affect, negative affect, loneliness, and relationship satisfaction were calculated via Pearson’s correlation coefficient.
2.6.7 Other analyses
We then carried out two-step structural equation modeling (SEM) in the AMOS pro-gram, using maximum likelihood estimation for parameter estimation. In this context, we first tested whether the measurement model was confirmed and then tested the structural model (Anderson & Gerbing, 1988; Kline, 2023). We also used the item parceling method (Bandalos, 2002; Nasser-Abu Alhija & Wisenbaker, 2006) for the unidimensional scales in the model we tested. A mediation analysis was conducted to examine the possible mediating role of negative affect and loneliness in the relationship between ghosting and relationship satisfaction. Additionally, we used bootstrap testing to determine whether the indirect effects are significant (Preacher & Hayes, 2008).”
The manuscript should specify the test used to assess normality (e.g., Kolmogorov-Smirnov test or Shapiro-Wilk test).
Authors’ response: We explained the following: “Normality assumptions were evaluated using the Shapiro-Wilk test, as it is more sensitive for smaller-moderate sample sizes; results indicated acceptable normality.”
Given the potential gender differences in the study variables, it would be insightful to conduct a t-test (Student’s t-test) comparing males and females. Have the authors considered this analysis?
Authors’ response: Thank you for this suggestion, we have performed the required analyses and reported results.
In the methods we explained the following: “Using data from Phase I and Phase II combined an independent samples t-test was conducted to compare ghosting scores between female (n = 512) and male (n = 251) participants. The test was used to determine if there was a statistically significant difference between the mean scores of the two groups.”
In the results we explained the following: “The analysis of independent samples t-test revealed virtually identical mean ghosting scores between females (M = 18.13, SD = 6.25) and males (M = 18.12, SD = 6.29). The independent samples I-test indicated no statistically significant difference between groups (t(761) = .037, p = .975, Cohen's d = .00). The mean difference was minimal at 0.01 points, with a standard error of difference of .48. These results demon-strate a complete absence of gender-based performance differences in the ghosting measure.”
Clarity and Formatting
Line 240 is unclear and should be revised for better readability.
Authors’ response: Line 240 (now line 319) was revised for clarify and formatting. Now it read as follows: “Accordingly, while there was a significant negative relationship between ghosting and conscientiousness (r = -.225, p < .001), and emotional stability (r = -.207, p < .001) personal-ity traits, there was no significant relationship between ghosting and extraversion (r = -.065, p > .05), agreeableness (r = -.078, p > .05), or openness to experience (r = -.009, p > .05) personality traits”.
Tables should emphasize statistical significance more clearly. This could be achieved by using symbols (e.g., an asterisk for significant values) or applying a light gray shading to significant results.
Authors’ response: We have added an asterisk * to flag significant values.
Discussion
The manuscript currently states that further research is warranted. However, specific suggestions for future research directions would enhance the conclusion. Concrete examples of potential studies would be valuable.
Authors’ response: We added the following: “We suggest the following areas to be explored. First, longitudinal designs could track ghosting experiences over time to establish causal pathways between ghosting, negative affect, loneliness, and relationship satisfaction, particularly during critical life transitions (e.g., emerging adulthood). Second, experimental studies could manipulate ghosting scenarios (e.g., simulated digital interactions) to isolate its immediate emotional and behavioral consequences. Third, comparative cross-cultural investigations are needed to explore how cultural norms (e.g., collectivism vs. individualism) shape perceptions and outcomes of ghosting, building on this Turkish adaptation. Fourth, mixed-methods approaches (e.g., qualitative interviews paired with physiological stress measures) could uncover motivations behind ghosting behaviors and their somatic impacts. Finally, intervention studies could evaluate strategies (e.g., resilience training, digital etiquette programs) to mitigate ghosting-related distress. These directions would deepen theoretical insights while in-forming practical solutions for individuals and communities navigating modern relational challenges.”
One of the most compelling sections of the discussion is the examination of the serial mediation role of negative affect and loneliness in the relationship between ghosting and relationship satisfaction (lines 333-339). This reasoning could be expanded by incorporating the role of emotional reasoning. To support this argument, I recommend citing Gangemi et al. (2021) Emotional Reasoning and Psychopathology, Brain Sciences, 11, 471. This study describes how emotions are not just reactions but can shape and bias judgment processes. This means that when someone experiences negative affect due to ghosting, they might not only feel bad but also interpret their situation in a more negative and self-defeating way. This can lead to increased loneliness, which in turn lowers relationship satisfaction. The cycle then reinforces itself:
Ghosting leads to negative affect (e.g., sadness, frustration).
Negative affect distorts perception—the individual might overgeneralize (e.g., “No one cares about me”), catastrophize (e.g., “I will never find a fulfilling relationship”), or personalize (e.g., “It must be my fault”).
This negative perception deepens loneliness—because the person withdraws or feels unworthy of relationships.
Loneliness further lowers relationship satisfaction, feeding back into negative emotions and continuing the cycle.
By incorporating emotional reasoning as a theoretical framework, the authors can better explain why negative affect and loneliness act as mediators. Instead of just showing that these variables are linked, this perspective provides a mechanistic explanation: people use their emotions as a heuristic, leading them to judge their relationships (and themselves) negatively, which exacerbates their distress."
Authors’ response: We that the reviewer for this important suggestion. After reading paper by Gangemi A, Dahò M, Mancini F. Emotional Reasoning and Psychopathology. Brain Sci. 2021 Apr 8;11(4):471. doi: 10.3390/brainsci11040471. PMID: 33917791; PMCID: PMC8068126
We added the following: “The serial mediation pathway identified in this study—ghosting → negative affect → loneliness → relationship satisfaction—can be clarified through emotional reasoning (Gangemi et al., 2021). Emotions shape cognitive appraisals and amplify psychological distress. For instance, individuals experiencing negative affect after ghosting may engage in distorted interpretations, such as overgeneralization (“No one values me”), catastrophizing (“I’ll never connect with others”), or personalization (“I caused this rejection”). These maladaptive cognitions intensify feelings of alienation and withdrawal, exacerbating loneliness. Loneliness then diminishes relationship satisfaction, reinforcing negative self-schemas and emotional distress. Gangemi et al.’s (2021) framework highlights how emotional reasoning connects affective states and cognitive biases, explaining the observed mediation. Future research could measure these cognitive distortions to clarify their role in the ghosting-loneliness-satisfaction cycle, especially in cross-cultural contexts with differing norms around emotional expression.”
